# Locally Grown Crops and Immunocastration in Fattening Heavy Pigs: Effects on Performance and Welfare

**DOI:** 10.3390/ani12131629

**Published:** 2022-06-24

**Authors:** Immaculada Argemí-Armengol, Daniel Villalba, Laura Vall, Ramon Coma, Josep Roma, Javier Álvarez-Rodríguez

**Affiliations:** 1Department of Animal Science, University of Lleida, 25198 Lleida, Spain; daniel.villalba@udl.cat (D.V.); lauravall9@gmail.com (L.V.); ramoncb694@gmail.com (R.C.); javier.alvarez@udl.cat (J.Á.-R.); 2Department of Environment, Matadero Frigorífico, 08279 Avinyó, Spain; jroma@avinyo.net

**Keywords:** immunocastrated male, heavy pig, field pea, protease inhibitors, feeding behaviour

## Abstract

**Simple Summary:**

Soybean meal is the most common protein source for feeding pigs, but the livestock sector relies on imported soybean. Some studies have shown no adverse effects of replacing soybean meal with alternative protein sources on growth performance of growing–finishing pigs. Thereby, measures to promote locally grown protein crops for animal feed are sought. Surgical castration of males is performed to avoid boar taint in meat and to prevent aggressive behaviour, although it is associated with pain for piglets, whereas immunocastration only requires two doses of a vaccine that transiently suppresses testicular functions. Moreover, few studies comparing feeding strategies and consumption habits according to different castration methods have been published. In the current study, the local pea diet, in nutritionally balanced diets, impaired the average daily gain and feed conversion ratio above body-weights of 110 kg, but not earlier, while increased passive behaviour, with less presence of tail biting. In immunocastrates, performance results were better than those in barrows, although dressing percentage decreased, and, after second vaccination, they switched from a boar- to a barrow-like status.

**Abstract:**

This study aimed to explore dietary replacement soybean meal (SBM) with local pea seeds (PS-L) and the effects of surgically castrated (CM) or immunocastration (IM) in heavy male pigs, on growth performance, feeding behaviour, and tail and ear lesions. Four treatments were arranged factorially (2 × 2), with two sexes and two dietary treatments (96 pigs in eight pens). The inclusion of PS-L was 25%, 30%, and 40% during three phases (40–80 kg, 80–110 kg, and 110–140 kg, respectively). No difference in average daily feed intake (ADFI) and body-weight (BW) between PS-L and SBM could be demonstrated (*p* > 0.05), but PS-L diet decreased the average daily gain (ADG) at 110–140 kg of BW (*p* < 0.001) and increased feed conversion rate (FCR) (*p* < 0.05). The ADG was higher (*p* < 0.01) in IM than CM in all fattening periods, and the FCR in IM was lower (*p* < 0.05) than in barrows. IM pigs had lower dressing percentage than CM (*p* < 0.01). Pigs fed a PS-L diet ate faster but increased their passive behaviour compared with those fed SBM. In conclusion, the PS-L diet did not reduce BW and improved passive behaviour, and IM grew more efficiently, regardless of diet.

## 1. Introduction

The production (part or all) of the feed components on a farm, and/or use of locally produced food components, can greatly reduce the environmental impact due to transport abroad [1] and improve the economic efficiency. Soybean meal (SBM) is currently the most common protein source for pigs’ diets in the European Union and in non-European countries, although it is mostly produced in South America [2]. Enhancing the local protein crop production may replace, or reduce reliance on, imported SBM [3]. There are thousands of legumes species that can grow around the world, such as beans and peas, in Europe, but which are characterised by a lower total protein and methionine content. Peas contains antinutritional factors such as protease inhibitors (as trypsin) and phenols (as tannins), which limit the nutritive value of legume-seed protein, although its white flowering varieties contain less antinutritional factors than coloured varieties [4]. Moreover, it is known that the use of specific percentages of peas in pig diets may not have negative effects on pig performance [5,6]. Prandini et al. [6] also found faba bean and a pea diet resulted in higher pork carcass weight than a soya diet. 

The dry-cured pork industry demands pigs having heavy weights (to achieve the maximum carcass weight and to increase quality traits), requiring greater age at slaughter and castration to avoid boar taint. Surgical castration of male piglets is common practice to avoid boar taint in pork and to prevent aggressive behaviour in heavier and older pigs, which has an impact on the meat and fat aroma. Castration of entire male pigs without pain relief is expected to be banned in the future, although it was previously scheduled for January 2018 [7]. Active immunisation against gonadotropin-releasing hormone (GnRH), so-called immunocastration, is an attractive alternative to surgical castration, and is now increasingly being used in many countries to reduce boar taint and aggressive behaviour. 

Moreover, immunocastration has a positive impact on feed conversion [8,9], and improves economic efficiency and the environmental impact [10]. The producer of the vaccine (Improvac^®^) recommends two doses, one after 10 weeks of age, and a second about 4 to 6 weeks before slaughter. The motivation to perform immunocastration in the finisher phase is to utilise the full growth potential of the entire male pigs until the second injection. The literature indicates that fat deposition increases from 4 weeks after the second vaccination [11]; however, even a shift in vaccination time from 4 to 6 weeks is relevant to calming animal behaviour and, thereby, to reducing restlessness in the barn [12]. However, some countries raise heavy pigs with a restricted feeding pattern, whose performance and eating patterns may differ from those of immunocastrated lighter pigs on an ad libitum feeding basis. Additionally, immunocastrated males may have different feeding patterns to those of entire males or castrated pigs; thus, it is reasonable to study feeding plans in this type of animal. 

Concerning tail lesions, tail biting is an abnormal behaviour, characterised by one pig’s dental manipulation of another pig’s tail [13]. Tail and ear biting have multi-factorial origins, such as the presence of slatted floors, a barren environment, competition for feed, inadequate feed intake, and other dietary reasons, in addition to heritability [14]. These behaviours may be boosted in heavy pigs under non-standard surgical castration. 

This research aimed to investigate the possibility of using local pea, and the impact of immunocastration, by replacing SBM in diet and surgical castration, on growth performance parameters of heavy pigs supplied feed ad libitum. Furthermore, the association between dietary treatment and sex on the behavioural patterns during feeding, and the occurrence of damaging behaviour (tail and ear biting) in different on-farm production stages, were evaluated.

## 2. Materials and Methods

Ethical review and approval were waived for this study, because all handling practices and experimental procedures were performed according to the Spanish Animal Protection Regulations RD 53/2013, which comply with European Union Directive 2010/63 concerning the protection of animals used for experimental and other scientific purposes.

### 2.1. Animals and Experimental Design

A group of ninety-six male pigs was used for this experiment, from a commercial pig herd (“El Mujal”, Navàs, Spain) originating from 37 Duroc dams sired by Berkshire boars, mated in the same batch. Forty-eight pigs were immunocastrated males (IM) at 20 weeks of age and forty-eight were surgically castrated males (CM) at <1 week of age. The pigs included in this experiment were individually ear-tagged (HDX quick transponders) at ten weeks of age (20.2 ± 0.41 kg body-weight, average ± standard error). The experiment was carried out in a commercial fattening barn, in the centre of Catalonia (“Granja El Graner”, Artés, Spain, at 316 m above sea level), where the same pen grouping was retained until a target body-weight at slaughter of 140 kg was reached. Allotment was performed by balancing IM and CM in each pen, and pigs were randomly allotted within similar BW groups. The experiment was conducted between December 2020 and May 2021. The environmental room temperature ranged between 13 and 23 °C. Pigs were housed in 8 pens with 50%-slatted floors (space allowance 1 m^2^/pig), and an automated feeding station (Compident Pig-MLP, from Schauer Agrotronic, Prambachkirchen, Austria) for monitoring daily individual feed intake and body-weight, and two nipple drinkers for water supply. The studied effects were castration method (immunocastration vs. surgical castration) and diet in terms of protein components (soya bean meal, import vs. field pea seeds, local). Each pen had 12 pigs, including 6 immunocastrated males and 6 castrated males, and four pens were used for each dietary treatment. 

From weaning to the onset of the study, all pigs received the same conventional diet (2309 kcal Net Energy [NE] and 10.0 g standardised ileal digestible [SID] lysine per kg). The study started when the pigs reached 15 weeks of age, with an average body-weight (BW) of 37.4 ± 1.16 kg for surgically castrated pigs and 39.0 ± 1.18 kg for immunocastrated males. The male pigs were immunocastrated with two subcutaneous injections (2 × 2 mL) of Improvac^®^ (Zoetis, Zaventem, Belgium); the first vaccination was at 20 weeks of age and the second at 24 weeks of age. Vaccination schedules and sampling are graphically represented in Figure 1.

### 2.2. Diets and Treatments 

Commercial sources of maize, barley, soybean meal (44% of crude protein), wheat and wheat bran, soy bean oil, and dehydrated alfalfa pellets were obtained for the control and experimental diets. Field white flowering peas were grown in an area close to the fattening farm, Avinyó (9 km away), and were harvested for the experiment (Table 1), whereas soybean meal was imported. 

Two dietary treatments were formulated for each phase: grower (40–80 kg), early finisher (80–110 kg), and late finisher (110–140 kg) in a feeding trial involving the control and experimental diet. The control diet (SBM), with no pea included, contained imported soybean meal at 19%, 13%, and 11% for grower, early finisher, and late finisher, respectively (Table 2). In the experimental diets (PS-L), local field pea seeds were gradually included at 25%, 30%, and 40%, for partial to total replacement of SBM (Table 2). 

Diets were formulated to be isonitrogen, isoenergy, and isoamino acidic for the first limiting indispensable amino acids. Inclusion of crystalline Lys was reduced, and inclusion of crystalline Met, Thr, and Trp was increased, as the concentration of field peas in the diets was increased, because pea protein contains less Met, Thr, and Trp than soybean protein. The proximate chemical composition of the pelleted compound feed was analysed according to the methods of the Association of Official Analytical Chemists [15]; total amino acids content and tryptophan are shown in Table 3. Total amino acids and individual proportions were analysed by liquid chromatography with fluorescence detector (HPLC-FLD), with the exception of tryptophan, which was analysed by liquid chromatography with a diode array detector (HPLC-UV (DAD)). The activity of trypsin inhibitors (TIA, mg/g) in local field pea was analysed according to the American Oil Chemists’ Society [13]. Each diet was available on an ad libitum basis for pigs: grower diets during the initial 49 days of the experiment, early finisher diets during the following 31 days, and late finisher diets during the final 36 days of the experiment; water was always available. Dietary schedules and sampling are graphically represented in Figure 1.

Therefore, four treatments resulted from the combination of two castration methods (immunocastrated males: IM vs. castrated males: CM) and two diets (control with soybean meal import: SBM vs. field pea seed local: PS-L). 

### 2.3. Performance and Feeding Behaviour Measurements

The automated pen feeding station (Compident Pig-MLP) allowed the pigs access to the feed throughout the whole day. A stationary scale allowed a simple width adjustment for animal weights from 25 to 150 kg. A gate placed in front of the trough prevented pigs from consuming another pig’s ration. The date and time of feeding, the time spent eating, and the weights of the feed consumed and left over by each individual pig were recorded. The leftovers were weighed automatically and offered to the next pig visiting the station. At the end of the experiment, daily data for feed consumption and body-weight (BW) for each animal were summarised, and the average daily feed intake (ADFI), BW gains, average daily gain (ADG), and feed conversion rate (FCR) were calculated within each phase (grower, early finisher, and late finisher). The individual BW was also recorded at the start of the experiment and at the end of each feeding phase with a gate weighing scale to verify the electronic feeding station recordings (correlation, r = 0.84).

Four behavioural traits were derived from the information obtained from the automated feeding station, after excluding visits where feed consumption was zero: number of visits per day, feeding time per visit (minutes), feed intake per visit (g), and feeding rate (g/minute).

### 2.4. Tail and Ear Lesions 

Tail biting is a parameter related to damage of the tail, ranging from superficial bites along the length of the tail to absence of the tail. Ear biting is defined as an interaction between two pigs, where the first performs a sustained mastication of a pen mate’s ear, which appears as wounds on the ears and blood. The eventual tail and ear lesions were measured at 18 weeks of age (≃50 kg, grower diet, before vaccination), at 23 weeks of age (≃80 kg, early finisher diet, after first vaccination), and at 26 weeks of age (≃110 kg, late finisher diet, after second vaccination). The assessor always maintained a distance of approximately 40 cm from the animal. Observation schedules and sampling are graphically represented in Figure 1.

Tail lesions were scored from 0 to 2 according to the Welfare Quality^®^ assessment protocol (www.welfarequalitynetwork.net/media/1018/pig_protocol.pdf, accessed on 30 November 2021), on a 3-point scale, where 0 = no evidence of tail biting or indication of superficial biting along the length of the tail, and no evidence of fresh blood or any swelling (red areas on the tail are not considered as wounds unless associated with fresh blood); 1 = evidence of chewing or puncture wounds with swelling; 2 = fresh blood is visible on the tail, there is evidence of some swelling and infection, and part of the tail tissue is missing and crust has formed. As the latter score was minimally observed, scores of 1 and 2 were grouped to simplify the categorical statistical assessment.

Ear lesions were scored using a scale from 0 to 1, where 0 = no lesion; 1 = lesions longer than 2 cm (evidence of bites/teeth marks with fresh blood and/or infection). The sum of scores on both left and right ears yielded a single score per pig per observation.

### 2.5. Abattoir Measurements 

The BW at slaughter was set at approximately 140 kg. Slaughtering was performed in a nearby commercial abattoir (Avinyó, 9 km from the farm) (Escorxador Frigorífic d’Avinyó S.A., Barcelona). Pigs were transported in the morning (between 7:00 and 8:00) with a truck having a relatively flat loading ramp. At the abattoir, animals were allowed to rest during a 4–5 h period with full access to water but not to feed. Pigs were stunned by CO_2_ (concentration 87%) using a dip lift system, exsanguinated, scalded, skinned, eviscerated according to standard commercial procedures, and split down the midline. Hot carcass weight was individually recorded before the carcasses were refrigerated in line processing at 2 °C.

### 2.6. Statistical Analysis 

The data were analysed with the JMP Pro 15 statistical software (SAS Institute, Cary, NC, USA). The experimental unit was the pen, whereas the animal was the observational unit. Thus, data for each animal were included in the model and analysed with a random regression coefficient mixed model. The average daily gain of every pen individual was obtained from the first derivative of body-weight on every plotted day. Similarly, the FCR was estimated from the first derivative of cumulative feed intake on BW throughout the growing–finishing period. The evolution of the different variables was modelled using a third-degree polynomial with each coefficient affected by dietary treatment (SBM vs. PS-L), method of castration (vaccination vs. surgical), and its interaction as fixed effects. As these interactions were never significant, they were excluded from the final model. Carcass weight and yield were analysed with treatment as a fixed factor and pen as a random factor. For carcass yield, the carcass weight was included as a covariate. Values are presented as least square means ± standard error of the mean (SE). The level of significance was set at 0.05. Differences between least square means were assessed with the Tukey test. The association between the dietary treatment and the ear and tail lesions (presence vs. absence) was assessed by contingency tables and a Pearson chi-square test. The level of significance was set at 0.05.

During the experiment, 1 pig died and 1 pig was discarded due to illness and injury, and their data were removed from the database; thus, the final dataset consisted of data from 94 pigs.

## 3. Results

The results are presented as main effects, as no significant interactions were found between the protein source and the method of castration.

### 3.1. Growth Performance

The effects of dietary treatment (SBM vs. PS-L) and the method of castration (IM vs. CM) on growth performance of pigs are shown in Table 4. 

At the onset of the study (at 15 weeks old), the pigs’ average body-weight (BW) was the same (*p* > 0.05), i.e., approximately 40 kg, in all treatments. The pea diet and the immunocastration did not affect the average BW at the end of three periods (at 21, 26, and 31 weeks of age), with slaughter weight being nearly 140 kg (*p* > 0.05). However, some growth characteristics were affected by the experimental diet and vaccination. 

During the three periods (40–80kg, 80–110kg, and 110–140 kg), the average daily feed intake (ADFI) was not affected by increasing the inclusion of field pea (25%, 30%, and 40%, respectively) (*p* > 0.05). Pigs fed the pea diet had the same average daily gain (ADG) and feed conversion rate (FCR) as pigs fed the control diet (*p* > 0.05) during the grower diet (40–80 kg, 25% pea) and the early finisher diet (80–110 kg, 30% pea). However, for the late finisher diet (110–140 kg), the pigs that received the pea diet (40% of pea) had lower ADG (*p* < 0.001) and worse FCR (*p* < 0.05) than those receiving the control SBM diet (0% pea).

From the beginning of the trial to day 49 (15 to 21 weeks old), and four days after the first Improvac^®^ vaccination, entire males ate less (*p* < 0.01) and grew equally to castrated males (CM), with better feed conversion efficiency (*p* < 0.05). There was a significant effect of immunocastration treatment on the growth rate of pigs, from day 49 to the slaughtering day (31 weeks of age), which grew faster (*p* < 0.01) than those in the control treatment (surgically castrated). Similarly, FCR of IM tended to be lower from 80 to 110 kg (26 weeks of age and seven days after the second Improvac^®^ vaccination), and was similar from 110 to 140 kg (31 weeks old) to CM. However, in the final period, until they were slaughtered, IM significantly increased their ADFI (*p* < 0.01). 

### 3.2. Carcass Outcomes

The effects of dietary treatment (SBM vs. PS-L) and the method of castration (IM vs. CM) on carcass outcomes of pigs are shown in Table 5. 

Carcass weight and carcass yield did not differ significantly (*p* > 0.05) between dietary treatments, i.e., SBM and PS-L diets. Although carcass weight was similar (*p* > 0.05) in IM and CM, dressing percentage was lowest in IM (*p* < 0.01).

### 3.3. Patterns in Feeding Behaviour 

The effects of dietary treatment (SBM vs. PS-L diets) and the method of castration (IM vs. CM) on the patterns of pig feeding behaviour, from 15 to 31 weeks old, are illustrated in Figure 2, Figure 3, Figure 4 and Figure 5.

For the overall trial, the pea diet and the immunisation against GnRF had no significant effect (*p* > 0.05) on the amount of feed consumed per feeding visit (g/visit, Figure 3a,b), or on the feed time spent per visit (min/visit, Figure 5a,b), in comparison to the control treatments. Feed intake per visit increased with increasing age by about 73%, from 163 g/visit, at 15 weeks old, to 281 g/visit, at 31 weeks old. Moreover, until 24 weeks of age, the time that pigs spent eating decreased with increasing days on feed, from 7.4 to 4.6 min/visit (Figure 3). However, at 24 weeks old (around 100 kg of BW), the pigs increased the time spent eating per visit until slaughtering, to 6 min/visit (Figure 5). 

The pea diet had no effect (*p* > 0.05) on the number of visits to the feed bowl per day, for either the grower diet (40–80 kg, 25% pea), the early finisher diet (80–110 kg, 30% pea), or the late finisher diet (110–140 kg, 40% pea), in comparison with the SBM (control) diet (Figure 2a). The vaccination of boars with the GnRH vaccine reduced the number of visits to the feed bowl until the second vaccination (*p* < 0.01), from 15 to 23 weeks old, and without any differences throughout the rest of fattening period, compared to surgical castration. Nevertheless, during the grower and early finisher diets, the amount of feed consumed per minute was higher in the pea diet than in the SBM diet (Figure 4a), 43 vs. 35 ± 2.7 g/min and 63 vs. 51 ± 3.5 g/min, respectively (*p* < 0.05); however, it did not differ in the late finisher diet (*p* > 0.05). The immunocastration had no effect on the pig eating rate (Figure 4b, *p* > 0.05). 

### 3.4. Tail and Ear Lesions 

Figure 6 and Figure 7 show the percentage of animals with tail and ear lesions according to the effect of dietary treatment (SBM vs. PS-L) during the three feeding phases: grower diet (40–80 kg, day 14 of trial), early finisher diet (80–110 kg, day 49 of trial and 22 weeks of age), and late finisher diet (110–140 kg, day 73 of trial and 26 weeks of age). Tail lesions (Figure 6) were observed in close to one-quarter of animals during the early finisher phase fed the control diet, and were more frequent than in the pea diet (*p* < 0.05). In the grower phase, there was no significant difference (*p* > 0.05) between diets, but in the finisher phase there was also a tendency for tail lesions to be more frequent in the SBM diet than in the PS-L diet (*p* < 0.10). Regarding ear lesions assessment, statistical analyses revealed no significant dietary treatment effect, although there was a tendency for higher ear lesions in SBM than in PS-L in the early finisher diet (Figure 7). 

## 4. Discussion

There were no significant interactions between feeding strategy and method of castration for growth rate or feed conversion rate, nor final BW or carcass yield, which evidenced that the protein source (PL-S vs. SBM) responded similarly to the surgical castration and immunocastration. A 40% inclusion of peas in the pelleted diet reduced the FCR from 110 kg of BW onwards, and immunocastration improved the FCR from 40 kg of BW, without any difference in final BW nor carcass weight. This suggests that it would be possible to reduce the input of SBM in pig diets and to apply immunocastration, thereby improving the sustainability of pig production.

Replacement of dietary soybean meal must be accomplished by formulating equivalent feed with equal net energy and standardised ileal digestible amino acids content [16,17], especially for lysine, methionine, threonine, and tryptophan. The similar performance between pigs fed the pea and soybean meal diets strongly indicates the viability of using temperate-grown legumes as potential alternatives to SBM [18]. Nevertheless, the variety of winter pea used in the present experiment had a lower crude protein content (18.6%) than the 20.6% published by FEDNA [19] for the spring varieties. According to FEDNA [20], a maximum inclusion rate of 200 g of peas/kg in balanced pig diets is suggested for pigs after 70 days of age. Based on the results reported here and elsewhere [6,18], grower and finisher pigs are able to tolerate a greater rate at up to 250 g/kg (grower, 40–80 kg), 300 g/kg (early finisher, 80–110 kg), and 400 g/kg (late finisher 110–140 kg), suggesting that the maximum dietary level can be higher from the biological viewpoint and that inclusion levels can be increased. Feeding the pea diet did not affect ADFI in any fattening period, and this observation indicates that the palatability of the feed is not influenced negatively by the inclusion of field peas in the diets, despite the potential astringent properties of their tannins that could decrease palatability [21]. In practice, condensed tannins are more commonly detected in coloured-flowered cultivars than in white-flowered cultivars [22], such as the winter white cultivar used in this experiment. The major anti-nutritional factor (ANF) in pea is trypsin, which may affect the voluntary feed intake by reducing amino acids digestibility, leading to reduced availability or imbalanced supply of amino acids to animals, and reducing voluntary feed intake [23]. According to Jezierny et al. [2], pea seeds contain rather low trypsin inhibitor activity (TIA), ranging from <0.2 to 5.0 mg TI/g, compared to those obtained in raw SBM (5.8 mg TI/g), even if most TI is destroyed during the desolventising–toasting stage of oil extraction. In our case, the trypsin inhibitor activity (TIA) of the peas was 3.9 mg TI/g, and could be classified as medium activity (3.7 to 5.3 mg/g) [24]. However, TIA up to 3.2 mg TI/g of diet did not affect pancreatic secretion of nitrogen or protein, or pancreatic chymotrypsin activity, in young pigs [25], and dietary levels of at least 4.7 mg TI/g of diet in growing pigs were tested without significant negative effects on performance criteria such as growth rate, feed intake, or feed conversion ratio [2]. Nevertheless, replacement of SBM with raw field pea in our trial, starting from 110 kg (late finishing phases), reduced the ADG and FCR, but did not affect ADFI and final BW at slaughtering. Therefore, our results suggest that the maximum dietary inclusion level based on economic evaluations may be lower than that deduced from biological restrictions. Auzins et al. [26] pointed out that local pea has the highest cost-efficiency for pig feeding and stands out compared to soybean cake, which has twice the cultivation and processing costs (costs per tonne of crop protein and amino acids, including ideal amino acid balance), in addition to advantages of short supply chains [27] and various environmental and climate benefits [28]. 

Moreover, with regard to feeding behaviour traits, pigs fed a pea diet had a higher feeding rate (g/min) than those fed a soyabean diet during the majority of the fattening period. In addition, the number of visits to the feeder in PS-L increased with increasing age. Consequently, pigs on the PL-S diet spent more time eating, and this fact may influence their more passive behaviour, with less presence of tail biting, compared to those on SBM. Finishing pigs try to adapt their feeding pattern to compensate for a nutrient restriction by, for example, increasing their feeding rate, which may reflect increased feeding motivation [29]. The same authors suggested that a change in feeding rate is not the only strategy that pigs adopt to reach their desired feed intake. Depending on the context, they may also modify the number of visits and the time spent feeding per visit. Furthermore, several studies revealed a complex, bidirectional communication between gut microbiota and the brain, affecting mood and behaviour, that is, the so-called microbiota–gut–brain axis (MGBA). Specifically, Kobek-Kjeldager et al. [30] suggested that diet-related risk factors for tail biting are under- and oversupply of protein precursors (including tryptophan), low dietary fibre content, and lack of satisfaction. Therefore, this response may be via gut microbial metabolites resulting in cytokine-induced sickness behaviour (i.e., immune activation) or shifting of the tryptophan-serotonin metabolism (associated with anxiety and depression). 

In the present study, the feeding pattern in barrows and immunocastrates may be affected by mixed housing in groups of 12 animals of both sexes (IM and CM). Consistent with Morales et al. [31], at the age of 15 weeks (≃ 40 kg), the ADFI was greater in castrated males than in entire males, and continued to be significantly higher until the second vaccination of Improvac^®^ (at 24 weeks of age). It was correlated with the number of visits to the food bowl per day, which was significantly higher in CM compared with IM (21 vs. 16 n/d, respectively), and the time that pigs spent eating per day (113 vs. 90 min/d, respectively). Prunier et al. [32] found a strong negative correlation between the reduction in testosterone and the increased feed intake, which explains why this occurred after the first immunisation against GnRH. Moreover, from the onset of the trial, castrated males grew slower than immunocastrated pigs (previous entire males) (until slaughter at 140 kg of BW, at the age of 31 weeks). Nutrient requirements are different before and after full immunisation [11], with a 5% to 8% higher lysine requirement in anti-GnRF immunised boars before the second immunisation, compared to physical castrates. This may suggest that, in this experiment, the diet had enough lysine (and the rest of the ideal amino acid balance) to meet the requirements of the growth of immunocastrated males, and surpassed the dietary requirements of surgically castrated males. Finally, our results agree with Dalla-Costa et al. [33], that found immunocastrated males had greater efficiency (FCR) than surgically castrated pigs, with no difference in final BW or carcass weight. However, IM presented lower carcass yield than CM, in agreement with other studies [34], which may be explained by greater weight of the testicles and the full intestinal tract. 

## 5. Conclusions

It can be concluded that an inclusion of at least 40% local pea seed in diets of crossbred pigs (≃140 kg of final BW) is biologically feasible and an alternative to imported soy, considering the lack of negative effects on final body-weight and carcass weight, and the increasing eating rate, which may counteract the occurrence of tail biting. Moreover, immunisation against GnRF (at 20 and 24 weeks old) improved the daily gain and feed conversion rate, compared to surgical castration during early life. Considering that no interaction resulted between dietary treatment and method of castration, local pea seeds may be included in diets for both immunocastrated and surgically castrated males.

## Figures and Tables

**Figure 1 animals-12-01629-f001:**
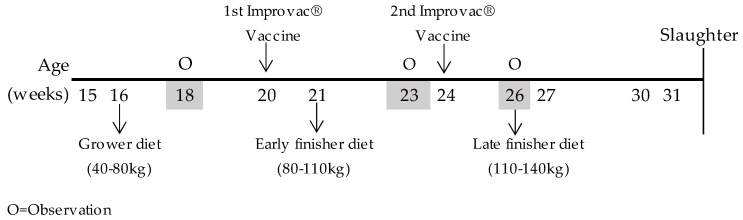
Time schedule for experimental design and sampling. O = observation (tails and ears).

**Figure 2 animals-12-01629-f002:**
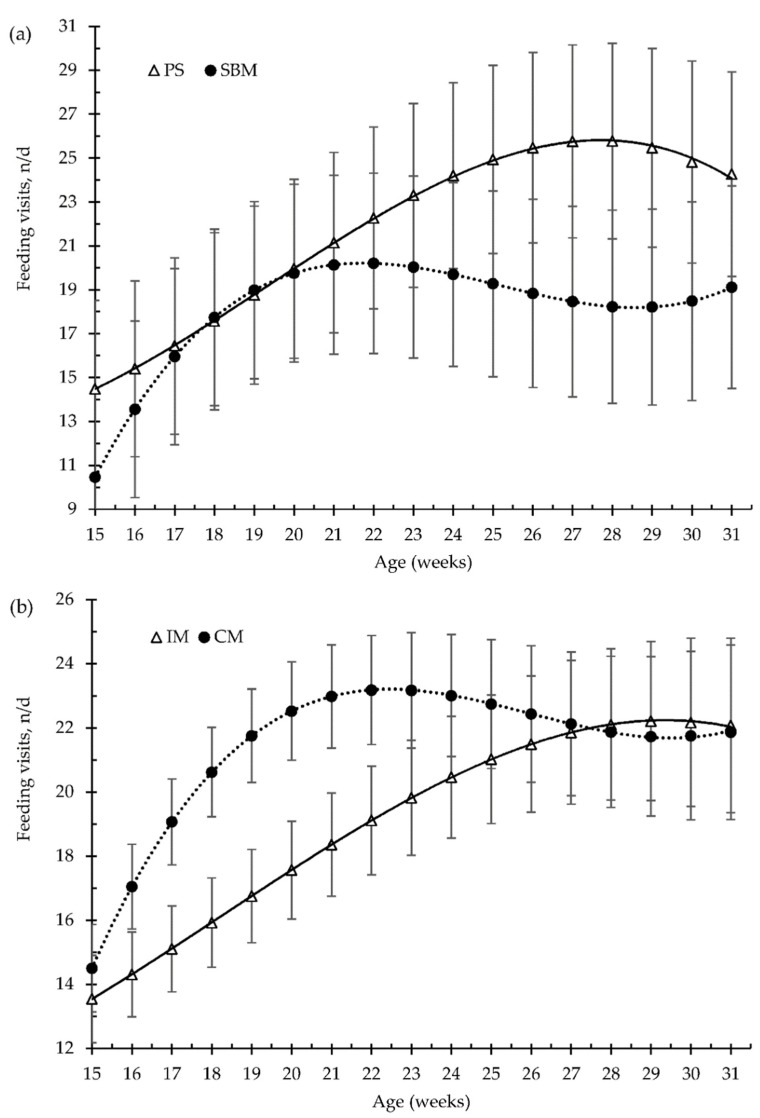
Growing pigs’ number of feeding visits (number per day, n/d) according to the feeding strategy (**a**) and the method of castration (**b**) with increasing age (weeks). SBM: soybean meal; PS-L: field pea seeds-local; IM: immunocastrated males; CM: castrated males.

**Figure 3 animals-12-01629-f003:**
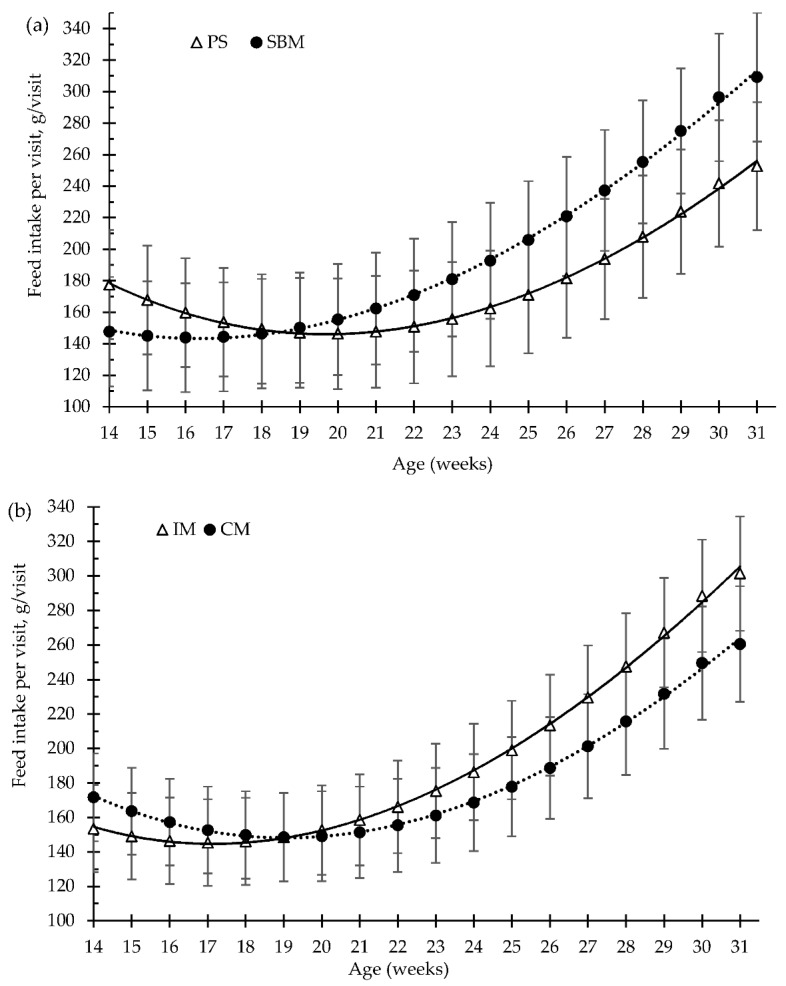
Growing pigs’ feed intake per visit (g/visit) according to the feeding strategy (**a**) and the method of castration (**b**) with increasing age (weeks). SBM: soybean meal; PS-L: field pea seeds-local; IM: immunocastrated males; CM: castrated males.

**Figure 4 animals-12-01629-f004:**
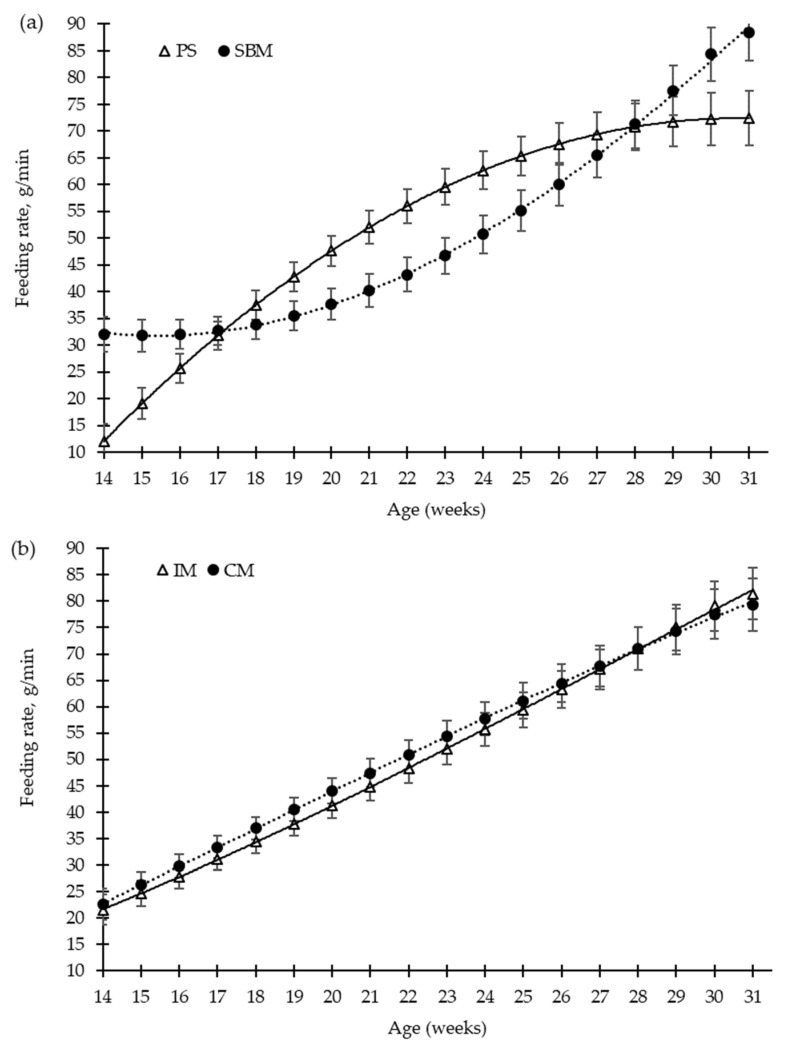
Growing pigs’ feeding rate (g/min) according to the feeding strategy (**a**) and the method of castration (**b**) with increasing age (weeks). SBM: soybean meal; PS-L: field pea seeds-local; IM: immunocastrated males; CM: castrated males.

**Figure 5 animals-12-01629-f005:**
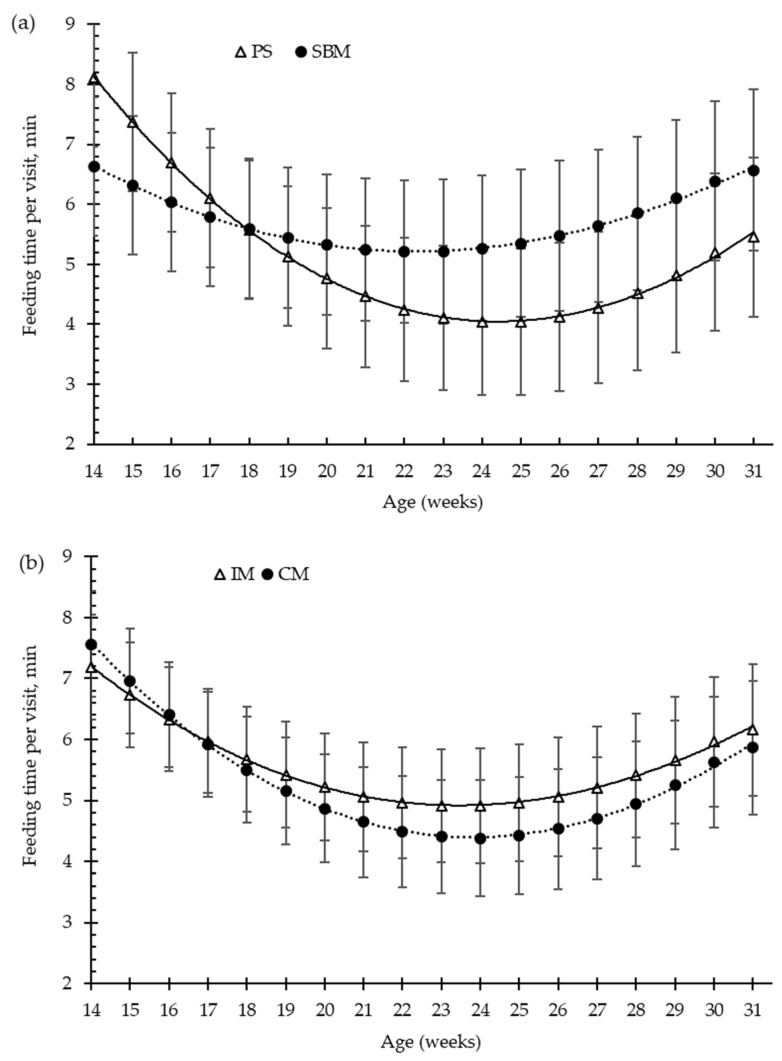
Growing pigs’ feeding time per visit (minutes, min) according to the feeding strategy (**a**) and the method of castration (**b**) with increasing age (weeks). SBM: soybean meal; PS-L: field pea seeds-local; IM: immunocastrated males; CM: castrated males.

**Figure 6 animals-12-01629-f006:**
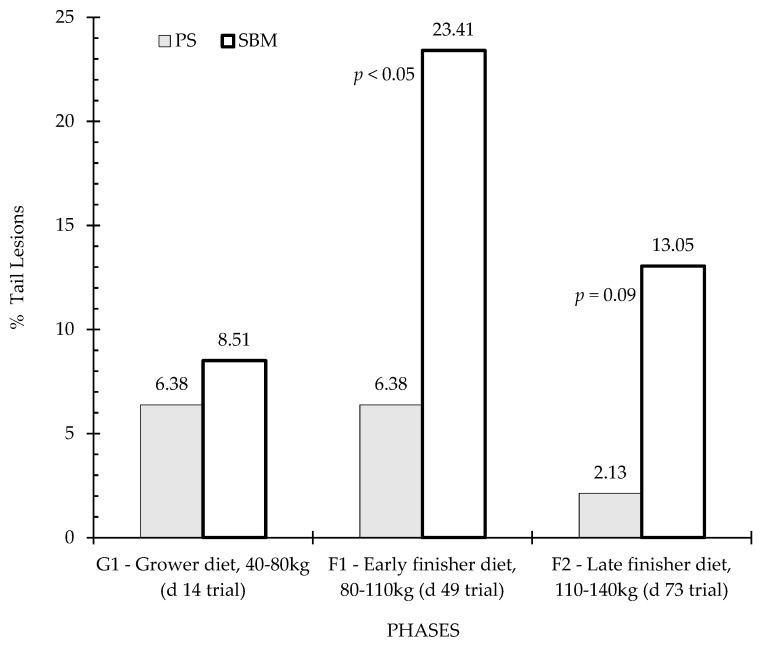
Percentage of tail lesions by feeding strategy. Columns represent the data of animals in three observation periods: P1—period before vaccination, P2—one week after first vaccination, and P3—three weeks after second vaccination. Differences are annotated with p-values in the graphs. SBM: soybean meal; PS-L: field pea seeds-local.

**Figure 7 animals-12-01629-f007:**
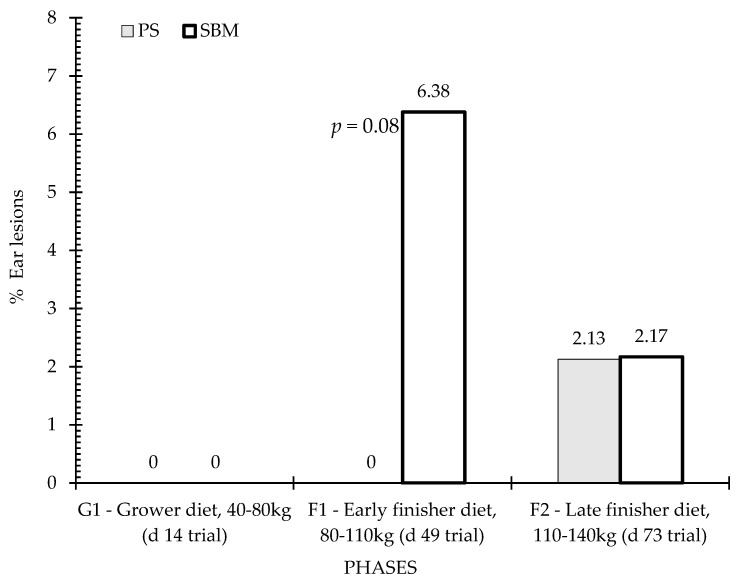
Percentage of ear lesions by feeding strategy. Columns represent the data of animals in three observation periods: P1—period before, P2—one week after first vaccination, and P3—five weeks after second vaccination. Differences are annotated with p-values in the graphs. SBM: soybean meal; PS-L: field pea seeds-local.

**Table 1 animals-12-01629-t001:** Chemical composition (g/kg) and amino acid content of pea seed local protein (g per 100 g of pea).

Parameter	
Dry matter	11.5
Crude protein, %	18.6
Ether-extract, %	2.1
Crude ash, %	5.8
Starch, %	42.9
Acid-detergent fibre, %	7.5
Neutral-detergent fibre, %	22.0
Crude fibre, %	6.2
**Trypsin Protease Inhibitors (AOCS)**	
Trypsin inhibitor activity (TIA, mg/g)	3.9
**Amino Acids**	
Lysine, %	1.4
Methionine, %	0.2
Threonine, %	0.7
Tryptophan, %	0.2
Isoleucine, %	0.8
Valine, %	0.9
Histidine, %	0.4
Arginine, %	1.6
Aspartic acid, %	2.1
Glutamic acid, %	3.7
Serine, %	0.9
Tyrosine, %	0.7
Phenylalanine, %	1.0
Leucine, %	1.5
Hydroxyproline, %	<0.03
Proline, %	1.1

**Table 2 animals-12-01629-t002:** Ingredient composition of the different diets: soybean meal vs. field pea seeds-local (g/100 g of feed).

Item	Grower (40–80 kg)		Early Finisher (80–110 kg)		Late Finisher (110–140 kg)	
	0% (SBM)	25% (PS-L)	0% (SBM)	30% (PS-L)	0% (SBM)	40% (PS-L)
Barley	35	24.8	30	22.2	40	25.6
Maize	43	35	31.4	25	39	25
Soybean meal, 44%	18.9	12	12.5	4.7	10.8	-
Soybean oil	0.5	0.5	0.5	0.5	0.5	0.5
Wheat	-	-	14.9	7.0	-	-
Wheat bran	-	-	8	8	-	-
Dehydrated alfalfa pellets	-	-	-	-	7.7	6.6
Field peas	-	25	-	30	-	40
Calcium carbonate	1.1	1.2	1.406	1.418	0.9	1.4
Sodium chloride	0.44	0.443	0.45	0.45	0.5	0.5
Mineral and vitamin premix	0.4	0.4	0.4	0.4	0.4	0.4
Bicalcic phosphate, 78.5%	0.485	0.405	0.188	0.079	0.1	-
L-Lysine HCl (Lys)	0.11	0.02	0.171	0.05	0.147	-
DL-Methionine, 99% (Met)	0.018	0.028	0.027	0.05	0.01	0.04
L-Tryptophan, 98% (Trp)	0.005	0.017	0.005	0.022	-	0.01
L-Threonine, 98.5% (Thr)	0.018	0.028	0.04	0.048	-	0.01

SBM: soybean meal; PS-L: field pea seeds-local.

**Table 3 animals-12-01629-t003:** Analysed nutrient content of the diets and essential amino acid content (g per 100 g of feed, unless otherwise stated).

Parameter	Grower (40–80 kg)		Early Finisher (80–110 kg)		Late Finisher (110–140 kg)	
	0% (SBM)	25% (PS-L)	0% (SBM)	30% (PS-L)	0% (SBM)	40% (PS-L)
NE, kcal/kg (calculated)	2388	2388	2350	2350	2325	2325
Dry matter	88.0	88.1	89	89.3	89.0	89.3
Crude protein, %	14.9	14.7	13.2	13.1	13.6	13.3
Ether-extract, %	3.6	3.4	3.5	3.6	3.3	3.6
Ash, %	4.9	6.0	4.8	6.3	5.0	6.1
Starch, %	48.8	48.0	50.5	49.4	47.8	48.4
Crude fibre, %	4.4	4.4	4.1	5.2	5.2	5.7
NDF, %	12.1	12.1	12.2	13.8	13.8	15.4
ADF, %	6.2	6.5	5.0	5.1	6.0	7.4
**Amino acids**						
Lysine, %	0.94	0.91	0.71	0.70	0.75	0.68
Methionine, %	0.3	0.33	0.24	0.26	0.22	0.22
Threonine, %	0.64	0.62	0.52	0.53	0.49	0.48
Tryptophan, %	0.17	0.19	0.17	0.15	0.16	0.13
Isoleucine, %	0.66	0.63	0.55	0.55	0.55	0.51
Valine, %	0.78	0.77	0.57	0.58	0.62	0.60
Histidine, %	0.46	0.44	0.35	0.34	0.36	0.32
Arginine, %	1.06	1.1	0.81	0.84	0.79	0.81
Aspartic acid, %	1.51	1.5	1.07	1.14	1.19	1.11
Glutamic acid, %	3.05	2.85	2.55	2.55	2.47	2.37
Serine, %	0.74	0.7	0.56	0.57	0.60	0.52
Tyrosine, %	0.57	0.54	0.44	0.44	0.48	0.42
Phenylalanine, %	0.83	0.79	0.66	0.67	0.71	0.63
Leucine, %	1.24	1.16	0.93	0.93	1.04	0.95
Hydroxyproline, %	0.034	<0.030	<0.030	<0.030	0.05	0.04
Proline, %	0.97	0.86	0.91	0.9	0.98	0.82

SBM: soybean meal; PS-L: field pea seeds-local.

**Table 4 animals-12-01629-t004:** Growth and feed efficiency achieved according to the feeding strategies and the method of castration.

Parameter	Feeding Strategy (F)		Method of Castration (C)		SE	Level of Significance ^1^	
	SBM	PS-L	IM	CM		F	C
**Body-Weight (BW), kg**							
Initial, day (d) 0	38.7	39.8	38.4	40	2.2	0.66	0.40
Grower diet (40–80kg), d 49	81.1	83	80.3	83.8	2.4	0.50	0.12
Early finisher diet (80–110kg), d 80	108.8	109.5	108.6	109.8	2.8	0.81	0.63
Late finisher diet (110–140 kg), d 116	141.7	139.5	142.9	138.4	3.3	0.53	0.15
**Average Daily Gain (ADG), g/day**							
Grower diet (40–80 kg)	867	882	856	893	21.8	0.49	0.10
Early finisher diet (80–110 kg)	893	856	910	839	21.3	0.09	<0.01
Late finisher diet (110–140 kg)	914	834	954	793	21.7	<0.001	<0.0001
**Average Daily Feed Intake (ADFI), g/day**							
Initial, day (d) 0	1504	1631	1389	1745	117	0.51	<0.01
Grower diet (40–80 kg), d 49	2675	3007	2695	2987	125	0.12	0.02
Early finisher diet (80–110 kg), d 80	3273	3628	3455	3447	145	0.12	0.96
Late finisher diet (110–140 kg), d 116	3829	4108	4272	3665	178	0.30	<0.01
**Feed Conversion Ratio (FCR)**							
Grower diet, 40–80 kg	3.35	3.78	3.37	3.76	0.23	0.18	0.01
Early finisher diet, 80–110 kg	3.91	4.62	4.07	4.47	0.28	0.06	0.07
Late finisher diet, 110–140 kg	4.48	5.46	4.76	5.18	0.35	0.02	0.16

^1^ Values are presented as least square means ± standard error (SE). The level of significance was set at 0.05, but tendencies were noted if the level of significance was below 0.10. The interaction between F and C was not significant (*p* > 0.05). SBM: soybean meal; PS-L: field pea seeds-local; IM: immunocastrated males; CM: castrated males.

**Table 5 animals-12-01629-t005:** Carcass characteristics according to the feeding strategies and the method of castration.

	Feeding Strategy (F)		Method of Castration (C)		SE	Level of Significance ^1^	
	SBM	PS-L	IM	CM		F	C
Final body mass, kg	145.2	145.1	147.4	142.9	1.92	0.94	0.11
Carcass weight, kg	105.5	104.8	105.3	105.0	1.53	0.76	0.87
Carcass yield, %	72.7	72.3	71.5	73.4	0.46	0.54	<0.01

^1^ Values are presented as least square means ± standard error (SE). The level of significance was set at 0.05, but tendencies were noted if the level of significance was below 0.10. The interaction between F and C was not significant (*p* > 0.05). SBM: soybean meal; PS-L: field pea seeds-local; IM: immunocastrated males; CM: castrated males.

## Data Availability

The data presented in this study are available on request from the corresponding author.

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
