# Peer review of "Locally Grown Crops and Immunocastration in Fattening Heavy Pigs: Effects on Performance and Welfare"

_animals, 2022, doi:10.3390/ani12131629_

Round 1
Reviewer 1 Report
All requested changes have been made. There are grammatical errors that should be corrected before publication.
Author Response
Thank you very much for your comments. All the co-authors proof read again the manuscript to improve its comprehension. Some grammar and style amends have been incorporated into the revised version (colour tracked).

Reviewer 2 Report
The manuscript has been substantially improved. However, I am afraid that the authors have not fully understood my comment in point 4.
Considering that the immunocastrated and surgically castrated pigs were reared in the same pens and in the absence of individual behavioral observations that allow distinguishing between biters and victims, I do not consider it correct to report any conclusion on lesions depending on the castration method.
I, therefore, suggest deleting what is written in the Simple Summary on lines 22-24 (from the word although onwards) and in the discussion on lines 467-68 (Whilst ... 110 kg).
Similarly, I do not consider figures 6b and 7b useful (which create confusion), which, in my opinion, should be removed.
In table 5, it is necessary to specify the meaning of F (feeding) and S (sex) in the legend.
Author Response
The manuscript has been substantially improved. However, I am afraid that the authors have not fully understood my comment in point 4.
Considering that the immunocastrated and surgically castrated pigs were reared in the same pens and in the absence of individual behavioral observations that allow distinguishing between biters and victims, I do not consider it correct to report any conclusion on lesions depending on the castration method.
I, therefore, suggest deleting what is written in the Simple Summary on lines 22-24 (from the word although onwards) and in the discussion on lines 467-68 (Whilst ... 110 kg).
We agree with the reviewer and we have deleted the sentences in the Simple Summary and in the Discussion.
Similarly, I do not consider figures 6b and 7b useful (which create confusion), which, in my opinion, should be removed.
We agree with the reviewer and we have removed the figures and the corresponding discussion regarding these variables.
In table 5, it is necessary to specify the meaning of F (feeding) and S (sex) in the legend.
A wrong letter was used to identify castration method P-values (S instead of C). It has been amended and the meaning of F (feeding strategy) and C (method of castration) is reported correctly now in the column headings of each table.

Reviewer 3 Report
The study is, in the light of the necessity to reduce the environmental impacts of food animal production of high signifcance. Replacing imported soy bean meal by locally grown crops without negative effects on the animals' performance, is highly appreciated by those that acre for theenvironment and the climate. The study is well designed and understandably presented.
However, the English should be intensively proof-red - maybe a professional proof-reader can be hired.
Here are some examples of what needs inguisticly be improved:
Line 10 and 11: ...pigs which demand is mostly.... (a comma is missing and the relative clause doen't sound correct)
Line 12: ...soy bean meals.. (the plural is not correct - there is only soy bean meal)
Line 24: it must read "surgically castrated" istead of "surgical castrated"
Line Line 43: ...diets in THE European Union and IN non-European countries
Line 44/45: Enhancing the LOCAL (or DOMESTIC) protein crop production...
Line 52/53: ...faba bean and pea's diet resulted in animal with a higher carcass weight at slaughterhouse... (This must read ...at THE slaughterhouse, or AT SLAUGHTER). And why is it faba bean diet and pea's diet - why the apostrophe in pea's?)
Line 98/99: immunocastrated males is = (correctly) IM, but the CM is described as "surgically castrated" - this should read "surgically castrated MALES = CM
Line 504 to 507: The sentence is not consistend in termes of grammar.
Please, be aware that these corrections are only some examples. There are more langauge weaknesses than these in the manuscript. I.e.: there must be a general proof-reading of the entire text.
Author Response
The study is, in the light of the necessity to reduce the environmental impacts of food animal production of high signifcance. Replacing imported soy bean meal by locally grown crops without negative effects on the animals' performance, is highly appreciated by those that acre for the environment and the climate. The study is well designed and understandably presented.
However, the English should be intensively proof-red - maybe a professional proof-reader can be hired.
Here are some examples of what needs inguisticly be improved:
Line 10 and 11: ...pigs which demand is mostly.... (a comma is missing and the relative clause doen't sound correct)
Many thanks for your comment. We rewrote the sentence as follows: Soybean meal is the most common protein source for feeding pigs, but livestock sector relies on imported soybean.
Line 12: ...soy bean meals.. (the plural is not correct - there is only soy bean meal)
We deeply appreciate your kind suggestions. We corrected the spelling mistake.
Line 24: it must read "surgically castrated" istead of "surgical castrated"
We deeply appreciate your kind suggestions. We corrected the spelling mistake.
Line Line 43: ...diets in THE European Union and IN non-European countries
Many thanks for your comment. We rewrote the sentence as follows: Soybean meal (SBM) is currently the most common protein source for pigs’ diets in the European Union and in non-European countries, although it is mostly produced in South America.
Line 44/45: Enhancing the LOCAL (or DOMESTIC) protein crop production...
We agree. We added “local”
Line 52/53: ...faba bean and pea's diet resulted in animal with a higher carcass weight at slaughterhouse... (This must read ...at THE slaughterhouse, or AT SLAUGHTER). And why is it faba bean diet and pea's diet - why the apostrophe in pea's?)
We deeply appreciate your kind suggestions. We corrected the spelling mistake. We have deleted “’s” and we have written “At slaughter”.
Line 98/99: immunocastrated males is = (correctly) IM, but the CM is described as "surgically castrated" - this should read "surgically castrated MALES = CM.
We agree. We added “males”
Line 504 to 507: The sentence is not consistend in termes of grammar.
Many thanks for your comment. We rewrote the sentence as follows: “It can be concluded that an inclusion of at least 40% local pea seed in diets of cross-bred pigs (≃140 kg of final BW) is biologically feasible and an alternative to soy imported, considering the lack of negative effects on final body-weight and carcass weight, and the increasing eating rate, which may counteract the occurrence of tail biting.”
Please, be aware that these corrections are only some examples. There are more language weaknesses than these in the manuscript. I.e.: there must be a general proof-reading of the entire text.
All the co-authors proof read again the manuscript to improve its comprehension. Some grammar and style amends have been incorporated into the revised version (colour tracked). We hope that these revisions have improved the manuscript clarity and we deeply-thank to reviewers’ thoughtful recommendations. Please let us to know if there is any further improvement we can handle.

This manuscript is a resubmission of an earlier submission. The following is a list of the peer review reports and author responses from that submission.
Round 1
Reviewer 1 Report
Title- Use of local resources (unclear what a local resource is?)
Title- As alternatives (to what?)
Line 10- unclear- Causes European countries to be dependent on imports?
Why are immunocastration and alternative protein sources in the same paper? Connection needs to be made somewhere.
Line 35: this difference only tended to differ? Unclear
Abstract: Has no numbers. Unclear why the 2 things are being examined together in a 2 by 2 design. Was there a treatment effect? Was there an interaction between the factors? This should be clear from the abstract.
Introduction- Unclear how this applies and where in terms of the SBM dependency and where we would expect the findings of this study to be applicable. Europe is mentioned as is the Mediterranean. Why would someone not in these regions be interested in this information? Can it be more broadly applied?
Line 51- Why does the reader need to know about ANF and how does it apply in this situation? Does it mean animals don’t grow as fast?
Line 54- Which always has an impact- Unclear what this is referring to?
The paragraph on surgical and immunocastration is unclear how it relates to this study specifically and how it relates to different feeding methods. What do we know about the use of immunocastration on the outcomes of this study and what gaps does this study address? Same for the tail biting paragraph. How is this information important and related to the study question? Why would we expect to need to know about tail biting? Is there some information on how it is related to immunocastration and protein source? What about ear biting?
Line 43- remove “considerable” or change to an adverb
Line 44- Soybean meal is the most…Run on sentence rephrase
Line 47- Other legume species… Run on sentence- rephrase
Materials and Methods
Line 78- Replace “due to” with “because”
Line 186- “when one week old” needs to be closer to what it is modifying and not at the end of the sentence (pigs surgically castrated at 1 week of age)
Line 90- The experiment was conducted… Run on sentence. Rephrase into multiple sentences.
Line 132- According “to”
Line 151- What is a “lateral and central barrier”?
Line 153- Prevented pigs from consuming another pig’s ration
Line 162- What is (r=.84)?
Line 164- Number “of” visits
Line 171- Appearing wounds on the ear and blood- unclear
Line 177- Bullet points are not necessary- use subheading if needed
Line 203- on the other hand… Why was logistic regression or ordinal regression not used to analyze the ear and tail lesions? How did you account for multiple measures on the same animals? The interaction term?
Results- Overall, the interaction between the protein source and the method of castration is not reported but it should be given the 2 x 2 factorial design. Even if it is not significant. There seems to be an interaction between day of age and the protein source on many of the outcomes. Was this investigated? The results would be nice to see the magnitude of the effect reported for each treatment for each time point and the significance.
Figures 2-5 - Are the values LSM and SE?
Section 3.1- Growth performance- Was there an interaction between protein source and method of castration? Please report.
Line 223- observed either (p > 0.05) in the average weights- Not clear what this means
Line 224-was similar between dietary treatments- Vague, be more specific
Line 230- The FCR the early finisher diet continued- Unclear English
Line 234- At the beginning of the experiment (day 0), castrated males (CM)- Is this the feed consumed in 1 day? Unclear.
Line 243- “Castrate’s method” is unclear, method of castration would be correct
The paragraph starting on line 234 could be more clear how the FCR differed by method of castration and whether it was different in each period and whether it differed overall.
Line 247- “few” is vague. This sentence is also long.
Line 288- “until the slaughter between the treatments” this is incorrect. “between the treatments” must be closer to what it is modifying, or it is modifying the slaughter. It should be slaughter, not “the” slaughter.
Line 309- “respectively” is not clear what this is referring to.
Tail and ear lesions: Please explain why logistic regression was not used in order to examine the effect of each treatment and their interaction in one model for each outcome.
Line 313- These wounds were seen only in a small… This is vague. Omit or be more specific
Line 338- “but no with ear lesions”- but no ear lesions?
Discussion- It is still not resolved for the reader why these 2 factors are being examined at the same time. They are treated entirely separately for the entire manuscript. The interactions or lack of interaction is never reported. The ending of the discussion is very sudden. It is unclear where the “local pea” is local and where this information would be applicable.
Line 343- It was more of a measure of aggression than welfare. “iceberg indicator” should be removed or defined. Restating the objectives is not needed. Consider replacing with an assessment of the findings of the study.
Line 365- starting with “In fact” is not a clear sentence.
Line 367-369- Although only one pulse… What is a pulse? How are the tannins related to the study outcome?
Line 370- What is the relevance of TUI to the study outcomes
Line 391- The major metabolites mediating this response… What response? Satiation or tail biting?
Line 417- Unclear if these are the numbers from the referenced study or from the current study
Line 422- In contrast,… In contrast to what? The findings of this study or the other study references? This paragraph could be more clear how this study agrees or disagrees with the prior research
Line 432- The aggressive and.. Unclear
Line 427- the paragraph starting Lealiifano is unclear what the authors think are the overall impacts of diet and method of castration on the tail lesions.
Line 444- “slightly FCR” is vague, rephrase
Line 445- but may be a hindrance to avoid redirected behaviours as tail biting- Unclear, rephrase
Reviewer 2 Report
The manuscript, although it has interesting insights, in my opinion, presents considerable criticalities.
1) The introduction reports a series of statements, all scientifically correct but unrelated. It does not clearly define the purpose of the experiment, namely the achievement of sustainable pig production. The authors should best express the ethical goal achieved through using a local feed resource and immunocastration (benefits in terms of animal welfare also due to an almost painless procedure).
2) The use of diets containing peas supplemented with synthetic amino acids in the feeding of pigs is not new. Indeed, the authors have successfully tested levels higher than those suggested by the current literature. Still, the same authors downsize the use of high amounts of pea in light of the decrease in feed efficiency (lines 345-357).
3) As for the experimental design, it is not clear what were the criteria that determined the allotment of the animals to the different pens (bodyweight? Litter of origin?) and, above all, it is not clear why the single animal was used as the experimental unit instead of the pen. The animals indeed had a feeding system that made it possible to establish individual consumption; however, the pen remains the smallest entity assigned to a particular dietary treatment. Consequently, the pen represents the experimental unit. Furthermore, the experimental design provided for 4 replications (pens) for each treatment (lines 92-98), and precisely the presence of replications constitutes the element that allows separating the possible confounding effects of the pen from those due to the dietary treatment (so, why the pen has not to be used as the experimental unit?)
4) In presenting the results, I do not believe that what was stated about body lesions is correct. Since each pen contained both six surgically castrated males and six immunocastrated males (lines 97-98), in the absence of individual behavioural observations that allow distinguishing between immunocastrated and non-immunocastrated biters, it is not possible to state that "indicators of aggressive activity as tail biting remained higher in immunocastrated pigs… "(lines 21-22). The same applies to what is reported in lines 37-38 and 333-338.
5) The graphs relating to eating behaviour bring little information and weigh down the manuscript. Therefore, it is suggested to limit the description of these results to the text.
6) The manuscript lacks an interesting aspect that would bring a remarkably innovative character to the study, consisting of the description of the slaughter parameters, mainly referring to the immunocastrated subjects receiving the 2 diets.